# Inhibitory Effect of AB-PINACA, Indazole Carboxamide Synthetic Cannabinoid, on Human Major Drug-Metabolizing Enzymes and Transporters

**DOI:** 10.3390/pharmaceutics12111036

**Published:** 2020-10-29

**Authors:** Eun Jeong Park, Ria Park, Ji-Hyeon Jeon, Yong-Yeon Cho, Joo Young Lee, Han Chang Kang, Im-Sook Song, Hye Suk Lee

**Affiliations:** 1College of Pharmacy, The Catholic University of Korea, Bucheon 14662, Korea; enddl0818@catholic.ac.kr (E.J.P.); hyacinthy7@catholic.ac.kr (R.P.); yongyeon@catholic.ac.kr (Y.-Y.C.); joolee@catholic.ac.kr (J.Y.L.); hckang@catholic.ac.kr (H.C.K.); 2College of Pharmacy and Research Institute of Pharmaceutical Sciences, Kyungpook National University, Daegu 41566, Korea; kei7016@naver.com

**Keywords:** AB-PINACA, drug interaction, drug-metabolizing enzyme, drug transporter

## Abstract

Indazole carboxamide synthetic cannabinoid, AB-PINACA, has been placed into Schedule I of the Controlled Substances Act by the US Drug Enforcement Administration since 2015. Despite the possibility of AB-PINACA exposure in drug abusers, the interactions between AB-PINACA and drug-metabolizing enzymes and transporters that play crucial roles in the pharmacokinetics and efficacy of various substrate drugs have not been investigated. This study was performed to investigate the inhibitory effects of AB-PINACA on eight clinically important human major cytochrome P450s (CYPs) and six uridine 5′-diphospho-glucuronosyltransferases (UGT) in human liver microsomes and the activities of six solute carrier transporters and two efflux transporters in transporter-overexpressing cells. AB-PINACA reversibly inhibited the metabolic activities of CYP2C8 (***K**_i_*, 16.9 µM), CYP2C9 (***K**_i_*, 6.7 µM), and CYP2C19 (***K**_i_*, 16.1 µM) and the transport activity of OAT3 (***K**_i_*, 8.3 µM). It exhibited time-dependent inhibition on CYP3A4 (***K**_i_*, 17.6 µM; ***k**_inact_*, 0.04047 min^−1^). Other metabolizing enzymes and transporters such as CYP1A2, CYP2A6, CYP2B6, CYP2D6, UGT1A1, UGT1A3, UGT1A4, UGT1A6, UGT1A9, UGT2B7, OAT1, OATP1B1, OATP1B3, OCT1, OCT2, P-glycoprotein, and BCRP, exhibited only weak interactions with AB-PINACA. These data suggest that AB-PINACA can cause drug-drug interactions with CYP3A4 substrates but that the significance of drug interactions between AB-PINACA and CYP2C8, CYP2C9, CYP2C19, or OAT3 substrates should be interpreted carefully.

## 1. Introduction

Indazole carboxamide synthetic cannabinoid (AB-PINACA, *N*-[(2*S*)-1-amino-3-methyl-1-oxobutan-2-yl]-1-pentyl-1H-indazole-3-carboxamide) was originally developed by Pfizer Inc. as a synthetic cannabinoid receptor agonist [1]. It binds and activates human cannabinoid receptor type 1 (hCB1) or type 2 (hCB2) but exhibits 23.3- and 40.9-fold higher in vitro binding affinity to hCB1 and hCB2, respectively, compared with the natural phytocannabinoid Δ9-tetrahydrocannabinol (Δ9-THC) [2,3]. AB-PINACA had receptor binding affinity to hCB1 and hCB2 of 2.87 nM and 0.88 nM (as ***K**_i_* values), respectively [2]. AB-PINACA substituted fully and dose dependently for Δ9-THC. However, AB-PINACA has been encountered outside medical settings as a synthetic constituent of herbal smoking mixtures, and was one of the most prevalent psychoactive substances in the USA in 2014–2015 [4]. It has been placed into Schedule I of the Controlled Substances Act by the US Drug Enforcement Administration since 2015 [4].

Biotransformation of AB-PINACA in human hepatocytes, liver microsomes, and urine samples from human subjects exposed to AB-PINACA has been evaluated, and 26 metabolites were identified in these samples [2,4,5,6,7,8]. In metabolism studies using human liver microsomes, AB-PINACA was shown to undergo extensive metabolism, and among the metabolites identified, AB-PINACA carboxylic acid, amide hydrolysis metabolite produced by carboxylesterase 1 (CES1) and CES2, was shown to be a major metabolite [7]. AB-PINACA was further metabolized by hydroxylation at the pentyl moiety, indazole moiety, and butanone moiety, ketone formation, carboxylation on the pentyl group, glucuronidation, and the combinations in human liver preparations and urine samples, but the metabolizing enzymes have yet to be identified [4,8]. It has been suggested that AB-PINACA may interact with drug metabolizing enzymes in the human body.

The monitoring of AB-PINACA as a new psychoactive substance began in Japan in 2012, and this has expanded to 21 countries since 2014 using human plasma and urine samples [1,9]. AB-PINACA has been found in drug products and biological samples [10,11]. One study found that of 58 cases of suspected impaired driving related to drug poisoning, AB-PINACA was detected in the blood samples of 25 cases at concentrations of 1.8–127 nM [10]. Another study reported that 10-month-old girl exposed by chewing an AB-PINACA contaminated cigarette had serum AB-PINACA and AB-PINACA pentanoic acid levels of 42 ng/mL and 345 ng/mL, respectively [11]. Other than these forensic analyses of AB-PINACA and its major metabolites, information regarding the interactions of drug metabolizing enzymes and transporters is limited. The use of CES1 inhibitor (i.e., 100 μM benzil) abolished the formation of AB-PINACA carboxylic acid in human liver microsomes [7], suggesting the importance of CES1 in the metabolism and pharmacokinetics of AB-PINACA and that inhibition of CES1 could cause drug interactions with AB-PINACA as well as substrate drugs that are metabolized by CES1, such as methylphenidate, oseltamivir, and enalapril [7,12,13,14].

Many synthetic cannabinoid receptor agonists interact with drug-metabolizing enzymes and transporters [15,16,17,18,19]. AM-2201 has been reported to potently inhibit the metabolic activities of cytochrome P450 (CYP) 2C9, CYP3A4, uridine 5′-diphospho-glucuronosyltransferase (UGT) 1A3, and UGT2B7 with IC_50_ values of 3.9 μM, 4.0 μM, 4.3 μM, and 10.0 μM, respectively [15]. MAM-2201 potently inhibited the metabolic activities of CYP2C9, CYP3A4, and UGT1A3 with ***K**_i_* values of 5.6 μM, 5.4 μM and 5.0 µM, respectively [16]. EAM-2201 inhibited the metabolic activities of CYP2C8, CYP2C9, CYP2C19, and CYP3A4 in a time-dependent manner and yielded irreversible kinetic parameters; ***K**_i_*: 0.54 µM and ***k**_inact_*: 0.0633 min^−1^ for CYP2C8, ***K**_i_*: 3.0 µM and ***k**_inact_*: 0.0462 min^−1^ for CYP2C9, ***K**_i_*: 3.8 µM and ***k**_inact_*: 0.0264 min^−1^ for CYP2C19, ***K**_i_*: 4.1 µM and ***k**_inact_*: 0.0250 min^−1^ for CYP3A4. EAM-2201 competitively inhibited UGT1A3 with a ***K**_i_* value of 2.4 µM [17]. APINACA inhibited CYP3A4 in a time-dependent manner with a ***K**_i_* 4.5 µM and ***k**_inact_* 0.04686 min^−1^ and noncompetitively inhibited UGT1A9 with a ***K**_i_* value of 5.9 µM [18]. AB-FUBINACA inhibited CYP2B6, CYP2C8, CYP2C9, CYP2C19, CYP2D6 with ***K**_i_* values of 15.0 μM, 19.9 μM, 13.1 μM, 6.3 μM, and 20.8 μM, respectively [19]. Taken together, synthetic cannabinoids showed moderate inhibition on drug metabolizing enzymes including irreversible time-dependent inhibition although they inhibited different CYP and UGT enzymes depend on their molecular structure. Drug metabolizing enzymes and transporters play critical roles not only in the absorption, distribution, metabolism, and excretion of their substrate drugs but also in causing drug interactions with concomitantly administered substrate drugs [20,21]. Based on these clinic drug interaction issues, evaluation of the effects of drugs and new drug candidates on drug-metabolizing enzymes and transporters has been recommended [22,23].

Here we aimed to investigate the effects of AB-PINACA on eight major CYP enzymes (i.e., CYP1A2, CYP2A6, CYP2B6, CYP2C8, CYP2C9, CYP2C19, CYP2D6, and CYP3A4), six major UGT enzymes (i.e., UGT1A1, UGT1A3, UGT1A4, UGT1A6, UGT1A9, and UGT2B7) using human liver microsomes. We also aimed to investigate the effect of AB-PINACA on six clinically important solute carrier transporters (i.e., organic anion transporter (OAT)1, OAT3, organic cation transporter (OCT)1, OCT2, organic anion transporting polypeptide (OATP)1B1, and OATP1B3), and two efflux transporters (i.e., P-glycoprotein (P-gp) and breast cancer resistance protein (BCRP)) to evaluate AB-PINACA-mediated drug interactions.

## 2. Materials and Methods

### 2.1. Materials

AB-PINACA and midazolam were purchased from Cayman Chemical Company (Ann Arbor, MI, USA). Acetaminophen, *N*-acetylserotonin, alamethicin, chenodeoxycholic acid, coumarin, 7-hydroxycoumarin, mycophenolic acid, naloxone, naloxone 3-β-d-glucuronide, phenacetin, reduced β-nicotinamide adenine dinucleotide phosphate (NADPH), sulfaphenazole, trifluoperazine, Trizma base, uridine 5′-diphosphoglucuronoic acid (UDPGA), Hank’s balanced salt solution (HBSS), Dulbecco’s Modified Eagle Medium (DMEM), and sodium dodecyl sulfate (SDS) were obtained from Sigma-Aldrich (St. Louis, MO, USA). Ultrapooled human liver microsomes (150 donors, mixed gender), LLC-PK1-MDR1 cells stably expressing P-gp, LLC-PK1-mock cells, HEK293-OAT1, -OAT3, -OCT1, -OCT2, -OATP1B1, and -OATP1B3 cells (HEK293 cells transiently overexpressing respective transporters), HEK293-mock cells, [*S*]-mephenytoin, 4′-hydroxymephenytoin, bufuralol, 1′-hydroxybufuralol, d_9_-1′-hydroxybufuralol, 1′-hydroxymidazolam, *N*-desethylamodiaquine, 4′-hydroxydiclofenac, ^13^C_2_,^15^N-acetaminophen, medium 199, fetal bovine serum (FBS), collagen-coated 24-Transwell plates, and poly-d-lysine-coated 96-well plates were purchased from Corning Life Sciences (Woburn, MA, USA). ^3^H-Methyl-4-phenylpyridinium (2.9 TBq/mmol), ^3^H-para-aminohippuric acid (0.13 TBq/mmol), ^3^H-estrone-3-sulfate (2.12 TBq/mmol), ^3^H-estradiol-17β-d-glucuronide (2.22 TBq/mmol), ^3^H-digoxin (1.103 TBq/mmol), and Optiphase scintillation cocktail were purchased from PerkinElmer Inc. (Boston, MA, USA). SN-38 was a product of Santa Cruz Biotechnology (Dallas, TX, USA). SN-38 glucuronide, troglitazone, diclofenac, mycophenolic acid β-d-glucuronide, chenodeoxycholic acid 24-acyl-β-glucuronide, and *N*-acetylserotonin β-d-glucuronide were purchased from Toronto Research Chemicals (Toronto, ON, Canada). All other chemicals were of the highest quality available. LLC-PK1-BCRP cells (LLC-PK1 cells stably expressing BCRP) were obtained from Dr. A.H. Schinkel (Netherlands Cancer Institute, Amsterdam, the Netherlands). LC-MS grade acetonitrile, methanol, and water were the products of from Fisher Scientific Co. (Fair Lawn, NJ, USA).

### 2.2. Inhibitory Effects of AB-PINACA on Eight Major CYP Activities in Human Liver Microsomes

The effects of AB-PINACA on CYP1A2, CYP2A6, CYP2B6, CYP2C8, CYP2C9, CYP2C19, CYP2D6, and CYP3A4 activities were measured in ultrapooled human liver microsomes as described in our previous report [18]. Human liver microsomal mixture (100 μL) containing 50 mM KH_2_PO_4_ buffer (pH 7.4), ultrapooled human liver microsomes (0.2 mg/mL), 10 mM MgCl_2_, 1.0 mM NADPH, and two sets of a cocktail of eight CYP probes (set A: 50 μM phenacetin, 2.5 μM coumarin, 2.0 μM amodiaquine, 5 μM bufuralol, 10 μM diclofenac, 100 μM [*S*]-mephenytoin, and 2.5 μM midazolam; set B: 50 μM bupropion, acetonitrile 0.5% (*v/v*)), and AB-PINACA (0.1–100 μM) were incubated for 15 min at 37 °C in a shaking incubator (Jeio Tech, Seoul, Korea) with or without preincubation with NADPH (30 min). After 15 min, the reaction was quenched by adding 100 μL of ice-cold methanol containing internal standards (IS) (^13^C_2_,^15^N-acetaminophen and d_9_-1′-hydroxybufuralol), followed by centrifugation at 13,000× *g* for 8 min at 4 °C. Supernatant from A and B sets (50 μL each) were mixed and 5 μL of the mixed supernatants were analyzed by LC-MS/MS. IC_50_ values were calculated by mean values of triplicate assays.

The ***K**_i_* values and inhibition modes for the inhibition of AB-PINACA on CYP2C8-, CYP2C9-, and CYP2C19-catalyzed hydroxylation of amodiaquine, diclofenac, and [*S*]-mephenytoin were measured as described in our previous report [15]. Human liver microsomal mixture (100 μL) containing 50 mM potassium phosphate buffer (pH 7.4), ultrapooled human liver microsomes (0.2 mg/mL), 10 mM MgCl_2_, 1.0 mM NADPH, various concentrations of AB-PINACA (1–40 μM), and amodiaquine (1, 2, 4, and 8 μM) for CYP2C8, diclofenac for CYP2C9 (2, 5, 10, and 20 μM) or [*S*]-mephenytoin for CYP2C19 (20, 40, 80, and 160 μM) were incubated at 37 °C in a shaking incubator. After 15 min, the reaction was quenched by adding 100 μL of ice-cold methanol containing ^13^C_2_,^15^N-acetaminophen (IS) for CYP2C8 or d_9_-1′-hydroxybufuralol (IS) for CYP2C9 and CYP2C19, followed by centrifugation at 13,000× *g* for 8 min at 4 °C.

The kinetic parameters (***K**_i_* and ***k**_inact_*) for time- and concentration-dependent inhibition of AB-PINACA against CYP3A4 activity in ultrapooled human liver microsomes were evaluated according to our previous report [18]. Various concentrations of AB-PINACA (0–100 μM) were preincubated with NADPH and ultrapooled human liver microsomes (1 mg/mL) in potassium phosphate buffer (50 mM, pH 7.4) at 37 °C in a shaking incubator. At predetermined time (5, 10, 20, and 30 min), the aliquots (10 μL) of preincubation mixture were transferred to fresh tubes containing 90 μL reaction mixtures of 50 mM potassium phosphate buffer (pH 7.4), 10 mM MgCl_2_, 1 mM NADPH, 2 μM midazolam, and incubated for 10 min at 37 °C in a shaking incubator. The second reactions were stopped by adding 100 μL of ice-cold d_9_-1′-hydroxybufuralol in methanol (IS).

CYP metabolites were determined by selected reaction monitoring (SRM) mode of LC-MS/MS using an Agilent 1290 Infinity UPLC system coupled with Agilent 6495 triple quadrupole mass spectrometer (Agilent Technologies, Wilmington, DE, USA) as described in our previous report [19] and Appendix A. The linear ranges were 5–200 pmol for acetaminophen and 1′-hydroxymidazolam, 15–600 pmol for 7-hydroxycoumarin and *N*-desethylamodiaquine, 1–40 pmol for hydroxybupropion, 1–100 pmol for 4′-hydroxydiclofenac, 0.5–20 pmol for 4′-hydroxymephenytoin and 1′-hydroxybufuralol. The accuracy and relative standard deviation values for eight metabolites were 94.7–100.1% and 5.8–9.1%.

### 2.3. Inhibitory Effects of AB-PINACA on Six Major UGT Activities in Human Liver Microsomes

The effects of AB-PINACA on the UGT1A1, UGT1A3, UGT1A4, UGT1A6, UGT1A9, and UGT2B7 activities were measured in ultrapooled human liver microsomes [18]. Human liver microsomal mixture (100 μL) containing 50 mM Tris buffer (pH 7.4), ultrapooled human liver microsomes (0.2 mg/mL), 5 mM UDPGA, alamethicin (25 μg/mL), 10 mM MgCl_2_, AB-PINACA (0.1–100 μM), and two sets of a cocktail of six UGT probes (set A: 2 μM chenodeoxycholic acid, 0.5 μM SN-38, and 0.5 μM trifluoperazine; set B: 1 μM *N*-acetylserotonin, 0.2 μM mycophenolic acid, and 1 μM naloxone). The incubation was continued for 60 min at 37 °C in a shaking water bath after addition of UDPGA, and the reactions were terminated by adding 50 μL of ice-cold propofol glucuronide and meloxicam in acetonitrile (ISs). After centrifugation, aliquots from supernatant from set A and set B (50 μL each) were mixed and 5 μL of the mixed supernatants were analyzed by SRM mode of LC-MS/MS as described in our previous report [19] and Appendix A. The linear ranges were 1–300 pmol for *N*-acetylserotonin β-d-glucuronide, chenodeoxycholic acid 24-acyl-β-glucuronide, mycophenolic acid glucuronide, naloxone 3-β-d-glucuronide, and SN-38 glucuronide, and 4–1200 pmol for trifluoperazine glucuronide. The accuracy and relative standard deviation values for six metabolites were 98.0–102.8% and 6.4–9.7%.

### 2.4. Inhibitory Effects of AB-PINACA on the Transport Activities of OCTs, OATs, OATPs, P-gp, and BCRP

HEK293-OCT1, -OCT2, -OAT1, -OAT3, -OATP1B1, and –OATP1B3 cells and HEK293-mock cells were seeded at a density of 10^5^ cells/well in poly-d-lysine-coated 96-well plates and cultured in DMEM supplemented with 10% FBS, 5 mM nonessential amino acids, and 2 mM sodium butyrate for 24 h at 37 °C in a humidified atmosphere of 8% CO_2_. Twenty-four hours after seeding of the cells, the growth medium was discarded and the attached cells were washed with prewarmed HBSS and preincubated for 10 min in prewarmed HBSS at 37 °C.

To examine the effects of AB-PINACA on the transport activity of OCT1 and OCT2, the uptake of 0.1 μM ^3^H-methyl-4-phenylpyridinium into HEK293-OCT1, -OCT2, and -mock cells was measured in the presence of AB-PINACA (0–250 μM) for 5 min. The uptake of 0.1 μM ^3^H-para-aminohippuric acid in HEK293-OAT1 and -mock cells, 0.1 μM ^3^H-estrone-3-sulfate in HEK293-OAT3, -OATP1B1, and -mock cells, and 0.1 μM ^3^H-estradiol-17β-d-glucuronide in HEK293-OATP1B3, and -mock cells was measured in the presence of AB-PINACA (0–250 μM) for 5 min. After 5 min, the incubation medium was discarded and the attached cells were washed three times with ice-cold HBSS (200 μL each) and lysed using 10% SDS solution (50 μL each). Cell lysates were mixed thoroughly with Optiphase scintillation cocktail (250 μL each) and the radioactivities in the cocktail mixtures were measured using a liquid scintillation counter (PerkinElmer Inc., Boston, MA, USA). The transporter mediated uptake of probe substrate was calculated by subtracting the uptake in HEK293-mock cells from the uptake in HEK293 cells overexpressing OCT1, OCT2, OAT1, OAT3, OATP1B1, and OATP1B3 transporters [24].

LLC-PK1-MDR1, LLC-PK1-BCRP, and LLC-PK1-mock cells were grown in tissue culture flasks in medium 199 supplemented with 10% FBS, and 50 μg/mL of gentamycin. The cells were seeded at a density of 10^5^ cells/well onto the insert membrane of 24 well Transwell plates and cultured for 5 days in medium 199 supplemented with 10% FBS, and 50 μg/mL of gentamycin. After reaching TEER values of the cells over 450 Ω·cm^2^, the B to A transport of digoxin in LLC-PK1-MDR1 and LLC-PK1-mock cells was initiated by adding 0.8 mL of HBSS containing 0.1 μM ^3^H-digoxin and AB-PINACA (0–100 μM) to the basal side and by adding 0.4 mL of fresh pre-warmed HBSS to the apical side of the Transwell chamber. Every 15 min for 1 h, an aliquot of 0.3 mL was taken from the apical side of the Transwell chamber and volume loss in the apical side was compensated by addition of 0.3 mL of fresh pre-warmed HBSS. The B to A transport of estrone-3-sulfate in LLC-PK1-BCRP and LLC-PK1-mock cells was also measured using the same protocol in the presence of 0.1 μM [^3^H]estrone-3-sulfate and AB-PINACA (0–100 μM) on the basal side. Aliquots (100 μL) of transport samples were mixed with 200 μL of Optiphase scintillation cocktail. The radioactivity of the probe substrate in the cocktail mixture was measured using a liquid scintillation counter. The B to A transport of digoxin and ES mediated by MDR1 and BCRP, respectively, was calculated by subtracting the transport of probe substrate in LLC-PK1-mock cells from that in LLC-PK1-MDR1 and LLC-PK1-BCRP cells and the B to A transport rate of probe substrate was calculated from the slope of the B to A transport of probe substrate versus time graph [18,25].

### 2.5. Molecular Docking of AB-PINACA

Molecular docking analysis was performed in Discovery Studio 2018 program (Dassault Systems BIOVIA, San Diego, CA, USA). X-ray crystal structures of human CYP2C8 (PDB code: 2NNH), CYP2C9 (PDB code: 1R9O), CYP2C19 (PDB code: 4GQS), and CYP3A4 (PDB code: 2D6Z) were obtained from Research Collaboratory for Structural Bioinformatics Protein Data Bank (RCSB-PDB, https://www.rcsb.org/) and the crystal structure of AB-PINACA was obtained from PubChem (PubChem code: 71301472). The CYP structures and AB-PINACA were optimized for use with Discovery Studio 2018 program. The active sites were determined with reference to the PDB. For docking analysis at the active site, spherical binding sites were formed on CYP2C8, CYP2C9, CYP2C19, and CYP3A4. AB-PINACA was docked at the binding site through the CDOCKER protocol. After docking, the interaction of protein and ligand in the binding site was analyzed using the protein-ligand interaction tool. The number of poses per ligand was set to at least 10, and the lowest CDOCKER interaction energy was used. Other parameters were set at default values.

### 2.6. Data Analysis

To calculate IC_50_ values (half-maximal inhibitory concentrations) of AB-PINACA, the inhibition data were fitted to an inhibitory effect model using SigmaPlot (version 12.0; Systat Software Inc., San Jose, CA, USA). The values of the inhibition constants (***K**_i_*) of AB-PINACA and the mode of inhibition were calculated from Lineweaver–Burk and Dixon plots [26]. Parameters for time-dependent inhibition such as ***K**_i_* and ***k**_inact_* (maximal rate of enzyme inactivation) were determined using SigmaPlot version 12.0 [18].

## 3. Results

### 3.1. Inhibitory Effects of AB-PINACA on CYP and UGT Activities in Human Liver Microsomes

Inhibitory effects of AB-PINACA on eight CYPs were estimated in ultrapooled human liver microsomes. To determine whether the time dependency of CYP inhibition by AB-PINACA is involved, the inhibitory effects of AB-PINACA on the catalytic activities of eight CYPs were measured with or without NADPH preincubation for 30 min (Figure 1, Table 1). AB-PINACA inhibited CYP2C8-, CYP2C9-, and CYP2C19-catalyzed hydroxylation of amodiaquine, diclofenac, and [*S*]-mephenytoin with IC_50_ values of 32.5 μM, 17.4 μM, and 20.0 μM, respectively, in human liver microsomes, but 30-min preincubation of AB-PINACA with NADPH did not significantly change the inhibitory potential of AB-PINACA toward CYP2C8, CYP2C9, and CYP2C19. However, preincubation of AB-PINACA with NADPH resulted in a 2.7-fold IC_50_ shift of CYP3A4-catalyzed midazolam 1′-hydroxylation (66.4 μM without NADPH preincubation vs. 24.4 μM with NADPH preincubation, Figure 1, Table 1), suggesting that AB-PINACA is a time-dependent inhibitor of CYP3A4. However, AB-PINACA did not significantly inhibit the catalytic metabolic activities of CYP1A2, CYP2A6, CYP2B6, and CYP2D6 in ultrapooled human liver microsomes regardless of preincubation of AB-PINACA with NADPH (IC_50_ > 100 μM in all cases) (Figure 1, Table 1).

Next, we investigated the inhibitory effects of AB-PINACA on the catalytic activities of CYP2C8-catalyzed amodiaquine *N*-deethylation, CYP2C9-catalyzed diclofenac 4′-hydroxylation, and CYP2C19-mediated [*S*]-mephenytoin 4′-hydroxylation with different substrate concentrations and varying concentrations of AB-PINACA and the results were transformed into Lineweaver–Burk and Dixon plots to determine the inhibition mode of AB-PINACA on the catalytic activities of CYP2C8, CYP2C9, and CYP2C19 along with the ***K**_i_* values (Figure 2). Enzyme kinetic results on the inhibition of CYP2C8 and CYP2C19 activities revealed mixed inhibition with ***K**_i_* values of 16.9 μM and 6.7 μM, respectively (Figure 2A,C, Table 2). AB-PINACA competitively inhibited CYP2C9 activity with a ***K**_i_* value of 16.1 μM (Figure 2B, Table 2).

To characterize the time-dependent inhibition of CYP3A4 by AB-PINACA, the inactivation kinetics for the formation of 1′-hydroxymidazolam from midazolam in the presence of AB-PINACA were measured: ***K**_i_* and ***k**_inact_* values of AB-PINACA were 17.6 μM and 0.04047 min^−1^, respectively (Figure 3 and Table 2).

AB-PINACA had low inhibitory potential for the glucuronidation activities of UGT1A3, UGT1A6, and UGT1A9 with IC_50_ values of 84.1 μM, 94.0 μM, and 65.2 μM, respectively, and showed negligible inhibition of the glucuronidation activities of UGT1A1, UGT1A4, and UGT2B7 in human liver microsomes in the concentration range of 0.1–100 μM AB-PINACA (Figure 4).

### 3.2. Inhibitory Effect of AB-PINACA on Drug Transporters

The inhibitory effects of AB-PINACA on the eight major transporters were evaluated using mammalian cells overexpressing OCT1, OCT2, OAT1, OAT3, OATP1B1, OATP1B3, P-gp, and BCRP. AB-PINACA had low inhibitory potential for OCT1-mediated methyl-4-phenylpyridinium uptake (IC_50_ value, 136 µM). AB-PINACA significantly inhibited OAT3-mediated ES uptake in a concentration dependent manner with an IC_50_ value of 11.8 µM. AB-PINACA did not significantly inhibit the transport activities of OCT2, OAT1, OATP1B1, OATP1B3, P-gp, or BCRP in the concentration ranges tested (Figure 5).

Among the eight transporters tested, OCT1 and OAT3 transporters that were inhibited by AB-PINACA were further subjected to enzyme kinetic studies to determine the mode of inhibition and ***K**_i_* values. Lineweaver–Burk and Dixon plots of the inhibitory effect of AB-PINACA on the OCT1-mediated methyl-4-phenylpyridinium uptake revealed mixed inhibition with a ***K**_i_* value of 145.7 µM (Figure 6A and Table 2). AB-PINACA competitively inhibited OAT3-mediated ES uptake with a ***K**_i_* value of 8.3 µM (Figure 6B and Table 2).

### 3.3. Molecular Docking Analysis

Our previous studies demonstrated that AB-PINACA inhibited phase I enzymes, which enhance the substrate solubility by hydroxylation, including CYP2C8, CYP2C9, CYP2C19, and CYP3A4 with ***K**_i_* values of 16.9, 16.1, 6.7, and 17.6 µM, respectively (Table 2). To elucidate the molecular mechanisms underlying the inhibitory effects of AB-PINACA against CYP2C8, CYP2C9, CYP2C19, and CYP3A4, we conducted molecular docking studies using the Discovery Studio computational docking program. We downloaded active co-crystal structures of human CYP2C8 (PDB code: 2NNH) [27], CYP2C9 (PDB code: 1R9O) [28], CPY2C19 (PDB code: 4GQS) [29], and CYP3A4 (PDB code: 4D6Z) [30] with substrate (i.e., CYP2C8-9-cis-retinoic acid, CYP2C9-flurbiprofen, CYP2C19-(4-hydroxy-3,5-dimethylphenyl)(2-methyl-1-benzofuran-3-yl)methanone, and CYP3A4-tert-butyl{6-oxo-6-[(pyridin-3-ylmethyl)amino]-hexyl}carbamate from RCSB-PDB, and each of these structure was utilized for molecular docking with AB-PINACA (PubChem code: 71301472), which were downloaded from PubChem (https://pubchem.ncbi.nlm.nih.gov). Before docking, we removed each substrate from the CYP2C8, CYP2C9, CYP2C19, and CYP3A4 co-crystal structures, conducted structure optimization for refinement using the minimization protocol, and performed rigid- and auto-docking with standard protocol. The results yielded docking scores for AB-PINACA of −29.36, −26.99, −33.15, and −32.34 kcal/mol against CYP2C8, CYP2C9, CYP2C19, and CYP3A4, respectively (Table 3). AB-PINACA had a hydrogen bond with the heme group of CYP2C8, and hydrophobic interactions with the R groups of Ile113, Phe205, Leu208, Val296, and Val336 at the active pocket (Figure 7A,B and Table 3). CYP2C9 had a hydrogen bond at Arg108, and hydrophobic interactions at the R groups of Leu206, Phe114, Aka297, and Leu366, and the heme group for the interaction with AB-PINACA at the active pocket (Figure 7C,D and Table 3). Hydrogen bonds of the R group of Asp239 and heme, and hydrophobic interactions with Leu102, Ala103, Phe114, and Phe476 in CYP2C19 at the active pocket, were involved in the interaction with AB-PINACA (Figure 7E,F and Table 3). Moreover, AB-PINACA formed multiple hydrogen bonds with the R group of Arg105 and heme, and the backbone chains of Arg372 and Glu374, and hydrophobic interactions at Phe108, Lle120, and Arg212 of CYP3A4 at the active pocket (Figure 7G,H and Table 3). These results suggest that AB-PINACA may inhibit phase I enzymes by binding to the active pocket of CYP2C8, CYP2C9, CYP2C19, and CYP3A4.

## 4. Discussions

AB-PINACA inhibited CYP3A4-catalyzed midazolam 1′-hydroxylation in a time-dependent manner (Figure 3). The inactivation efficiency of AB-FUBINACA on CYP3A4, calculated by dividing ***k**_inact_* (0.04047 min^−1^) by ***K**_i_* (17.6 μM) was 2.3 mL/μmol/min. Similar time-dependent inhibition of CYP3A4 has been reported for EAM-2201 (***k**_inact_*/***K**_i_*, 6.9 mL/μmol/min), a halogenated naphthoylindole synthetic cannabinoid [17], and APINACA (***k**_inact_*/***K**_i_*, 10.3 mL/μmol/min), an indazole carboxamide synthetic cannabinoid [18]. The calculated values of inactivation efficiency of AB-PINACA as well as the other synthetic cannabinoids, EAM-2201 and APINACA, were in the range of representative clinical CYP3A4-mediated drug interactions of clarithromycin (2.8 mL/μmol/min) and verapamil (11.2 mL/μmol/min) [31]. These findings suggest the possibility of CYP3A4-mediated drug interaction of AB-PINACA with CYP3A4 substrates, such as nifedipine, felodipine, simvastatin, atorvastatin, cyclosporine, clarithromycin, and midazolam [32].

However, AB-PINACA, an indazole carboxamide synthetic cannabinoid, inhibited CYP2C8-catalyzed amodiaquine *N*-deethylation (***K**_i_*, 16.9 µM), and CYP2C9-catalyzed diclofenac 4′-hydroxylation (***K**_i_*, 6.7 µM) with a mixed mode of inhibition (Table 2). It also competitively inhibited CYP2C19-mediated [S]-mephenytoin 4′-hydroxylation with a ***K**_i_* value of 16.1 µM (Table 2). As a result that the reported plasma concentration of AB-PINACA is in the range of 1.8–127 nM, although the dose regimen was not specified [10], and was much lower than the ***K**_i_* values of AB-PINACA, the inhibition of CYP2C8, CYP2C9, and CYP2C19 by the presence of AB-PINACA will not cause clinically significant drug–drug interactions.

Previous studies have reported that the active site cavity of CYP2C8 is located on either side of the helix B-C loop and is bound by helix I, the helix F-G region, portions of β-sheet 1, the turn in β-sheet 4, and the loop between helix K and β-sheet 1 [27]; therefore, it has sufficient space to capture AB-PINACA. However, unlike with CYP2C8-9-*cis*-retinoic acid crystal complex, in which 9-*cis*-retinoic acid formed hydrogen bonds with Arg241, the carboxamide group of AB-PINACA formed hydrogen bonds with heme (2.95 Å) of CYP2C8 (Figure 7A,B). Importantly, AB-PINACA shared the same amino acids, Ile113, Phe205, Leu208, Val296, and Val336 with 9-*cis*-retinoic acid to form hydrophobic interactions with CYP2C8, which formed additional interaction with Ile106 and Ile476 (Figure 7A,B and Table 3). As a result that CYP2C8 forms a homodimer that is connected by two molecules of palmitic acid and contains two 9-*cis*-retinoic acids in the active pocket in its crystal structure, it is not clear whether or not the inhibition of AB-PINACA against CYP2C8 requires two *cis*-retinoic acids.

As a result that human CYP2C9 and CYP2C19 have roughly 91% amino acid identity, it is predicted that the 3D structures and the substrate selectivity may be redundant. However, although published structures of these two enzymes do not show marked differences in the active site cavity, it is surprising that they exhibit distinct substrate selectivity [29,33,34]. This suggests that conformational changes underlie differences in the substrate selectivity [28]. Our study yielded docking scores of AB-PINACA against CYP2C9 and CYP2C19 of −26.99 and −33.15 kcal/mol, respectively (Table 3). The difference in binding energy may be due to the interactions of hydrogen bonds between AB-PINACA and Arg108 of CYP2C9 or Asp239/Heme of CYP2C19 (Figure 7C). Structural analyses between AB-PINACA and CYP2C9 or CYP2C19 revealed that the distance between AB-PINACA and Arg108 was 4.47 Å, whereas CYP2C19 had distances of 2.95 and 2.62 Å between AB-PINACA and Asp239 and Heme, respectively (Figure 7C–F). Moreover, our results revealed that AB-PINACA efficiently inhibited CYP2C19 activity with a lower ***K**_i_* value than CYP2C9 (Table 2). Additionally, CYP3A4 had diverse hydrogen bonds to Arg105, Arg372, Glu374, and Heme with 2.26–2.89 Å and energy score of −32.34 kcal/mol (Figure 7G,H and Table 3). Since interactions of CYP3A4 with tert-butyl{6-oxo-6-[(pyridin-3-ylmethyl)amino]hexyl}carbamate or midazolam, a CYP3A4 substrate, are known to occur through Arg106, Ala370, and Glu374, or Arg105 and Ile369, respectively [30], it is possible that the interaction between CYP3A4 and AB-PINACA is through the region making up the active site of CYP3A4 containing Arg105/106 and Ilu369-Glu374. These results suggest that AB-PINACA may preferentially interact with CYP2C19 at the active site under physiological conditions. However, the observation that the distances between AB-PINACA and amino acids of CYP3A4 are short for hydrogen bonds (Figure 7G,H and Table 3) means that crystallization of AB-PINACA and CYP3A4 may be necessary to resolve this issue.

We investigated the in vitro inhibitory effects of AB-PINACA, AB-FUBINACA, and APINACA, structural analogues of indazole carboxamide synthetic cannabinoids on the CYPs, UGTs, and transporter activities in this study and the previous reports [18,19]. APINACA (*N*-(1-adamantyl)-1-pentyl-1H-indazole-3-carboxamide) showed time-dependent inhibition on CYP3A4, which was similar to AB-PINACA. AB-FUBINACA (*N*-(1-amino-3-methyl-1-oxobutan-2-yl)-1-(4-fluorobenzyl)-1H-indazole-3-carboxamide) did not inhibited CYP3A4 but reversibly inhibited CYP2B6, CYP2C8, CYP2C9, CYP2C19, and CYP2D6. AB-PINCACA competitively inhibited CYP2C8, CYP2C9, and CYP2C19 in this study. We also elucidated that the pentyl group at indazole ring and amino-3-methyl-1-oxobutane moiety interact with the active sites of CYP2C8, CYP2C9, CYP2C19, and CYP3A4 through the hydrogen bonding and hydrophobic interaction via molecular docking results (Figure 7 and Table 3). Considering the similarity between AB-PINACA and APINACA (pentyl group at indazole ring) and difference with AB-FUBINACA (fluorobenzyl group at indazole ring), pentyl group at indazole ring seemed to be important for the time-dependent inhibition of CYP3A4. In addition, amino-3-methyl-1-oxobutane moiety seemed to be important for the inhibition of CYP2C8, CYP2C9, and CYP2C19 by AB-PINACA and AB-FUBINACA. We also reported that AB-PINACA inhibited OAT1 and OAT3 in this study. However, APINACA did not inhibit the transport activity of OCT1/2, OAT1/3, and OATP1B1/1B3 [18]. The structural differences between APINACA and AB-PINACA or AB-FUBINACA are the adamantyl moiety for APINACA and amino-3-methyl-1-oxobutane moiety for AB-PINACA. Our literature review revealed that ritonavir and ledipasvir, which have 2-amino-3-methyl-1-oxobutyl residue, have been reported as inhibitors of OCT1 and OCT2 [35,36,37]. Therefore, we speculated the amino-3-methyl-1-oxobutane moiety in AB-PINACA may have inhibited OCT1 transporter in this study. To date, there have been no reports regarding OAT3 inhibitors with an amino-3-methyl-1-oxobutane moiety. However, some organic cations have the capacity to interact with OAT transporters [38,39,40,41] and OAT3 has a propensity to bind with some cations that structurally overlap with OCT ligands [42]. For example, cimetidine (1-cyano-2-methyl-3-[2-[(5-methyl-1H-imidazol-4-yl)methylsulfanyl]ethyl] guanidine) and sitagliptin ((3R)-3-amino-1-[3-(trifluoromethyl)-6,8-dihydro-5H-[1,2,4]triazolo [4,3-a]pyrazin-7-yl]-4-(2,4,5-trifluorophenyl)butan-1-one) have been reported as inhibitors of both OCT1 and OAT3 [39,43,44]. Based on these previous reports, the amino-3-methyl-1-oxobutane moiety of AB-PINACA may explain the capability to inhibit both OCT1 and OAT3. However, to our knowledge, the capability for OCT and OAT inhibition by the adamantyl moiety has not been previously reported.

AB-PINACA inhibited OCT1-mediated methyl-4-phenylpyridinium uptake with a mixed mode of inhibition and also competitively inhibited OAT3-mediated estrone-3-sulfate uptake with ***K**_i_* values of 145.7 μM and 8.3 μM, respectively (Figure 6 and Table 2). Although the clinical data regarding the pharmacokinetics of AB-PINACA for prediction of AB-PINACA-induced drug interaction are limited, plasma concentration of AB-PINACA (1.8–127 nM) in humans much lower than the in vitro ***K**_i_* values of AB-PINACA for OCT1 and OAT3 is suspected for AB-PINACA intoxication [10]. Thus suggesting a low possibility of clinically significant drug–drug interaction between AB-PINACA and concomitantly administered OCT1 or OAT3 substrate drugs (i.e., metformin, cimetidine, pravastatin, diuretics, and non-steroidal anti-inflammatory drugs) [45]. However, other transporters, such as OCT2, OAT1, OATP1B1, OATP1B3, P-gp, and BCRP, interact poorly with AB-PINACA, even at high AB-PINACA concentrations (up to 250 μM). This suggests that AB-PINACA shows remote drug interaction likelihood for OCT2, OAT1, OATP1B1, OATP1B3, P-gp, and BCRP transporters.

## 5. Conclusions

The CYP, UGT, and transporter-mediated drug interaction potentials of AB-PINACA were evaluated based on in vitro inhibitory effects of AB-PINACA on major CYP and UGT enzyme activities and the transport activities of eight drug transporters. Clinically significant drug interaction potentials between AB-PINACA and CYP2C8, CYP2C9, CYP2C19, CYP3A4, or OAT3 substrates should be evaluated carefully, although the plasma concentrations of AB-PINACA reported in drug abusers were much lower than the ***K**_i_* values.

## Figures and Tables

**Figure 1 pharmaceutics-12-01036-f001:**
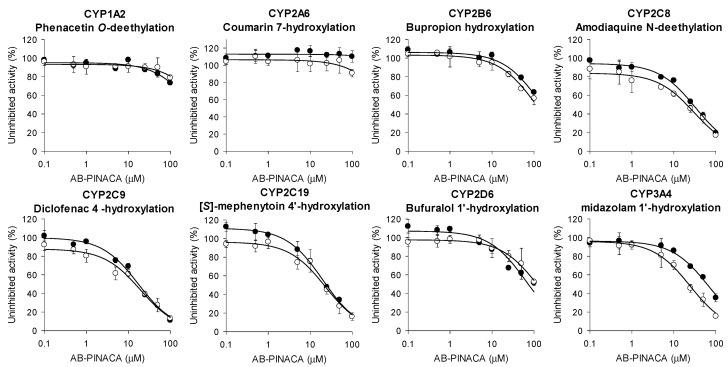
Inhibitory effects of AB-PINACA on metabolic activities of CYP1A2, CYP2A6, CYP2B6, CYP2C8, CYP2C9, CYP2C19, CYP2D6, and CYP3A4 in ultrapooled human liver microsomes with (◯) and without (⚫) 30-min preincubation with NADPH at 37 °C. The cocktail CYP probes consist of set A (50 μM phenacetin, 2.5 μM coumarin, 2.0 μM amodiaquine, 10 μM diclofenac, 100 μM [*S*]-mephenytoin, 5.0 μM bufuralol, and 2.5 μM midazolam) and set B (50 μM bupropion). Data are means ±SD (*n* = 3).

**Figure 2 pharmaceutics-12-01036-f002:**
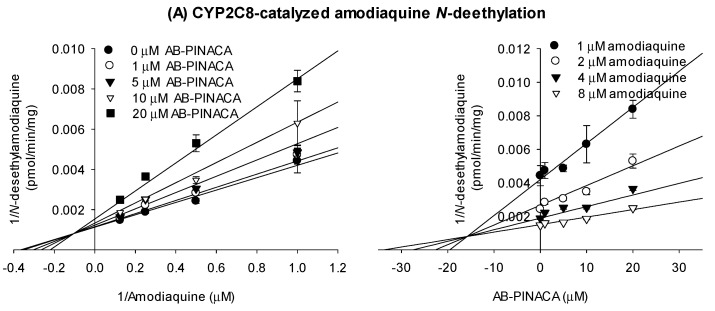
Lineweaver–Burk (left panel) and Dixon (right panel) plots for the inhibitory effects of AB-PINACA (1–40 μM) on (**A**) CYP2C8, (**B**) CYP2C9, and (**C**) CYP2C19 activities in the presence of varying concentrations of substrates, amodiaquine (1, 2, 4, and 8 μM), diclofenac (2, 5, 10, and 20 μM), or [*S*]-mephenytoin (20, 40, 80, and 160 μM) in ultrapooled human liver microsomes. Data are means ± SD (*n* = 3).

**Figure 3 pharmaceutics-12-01036-f003:**
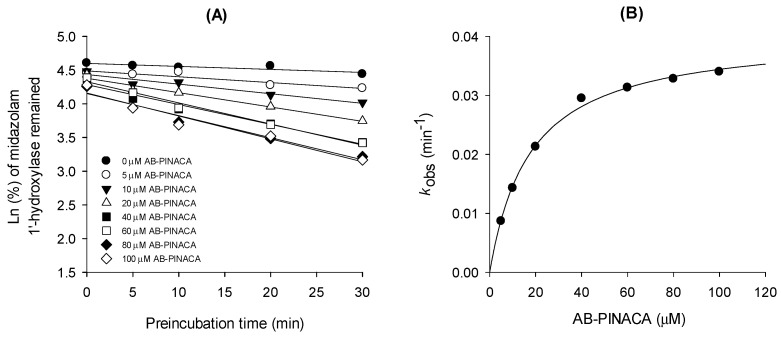
(**A**) Inactivation kinetics of midazolam 1′-hydroxylation by AB-PINACA (5–100 μM) in ultrapooled human liver microsomes and (**B**) the relationships between the ***K**_obs_* values and the AB-PINACA concentrations for the determination of the ***K**_i_* and ***k**_inact_* values for time-dependent inhibition of CYP3A4 activity.

**Figure 4 pharmaceutics-12-01036-f004:**
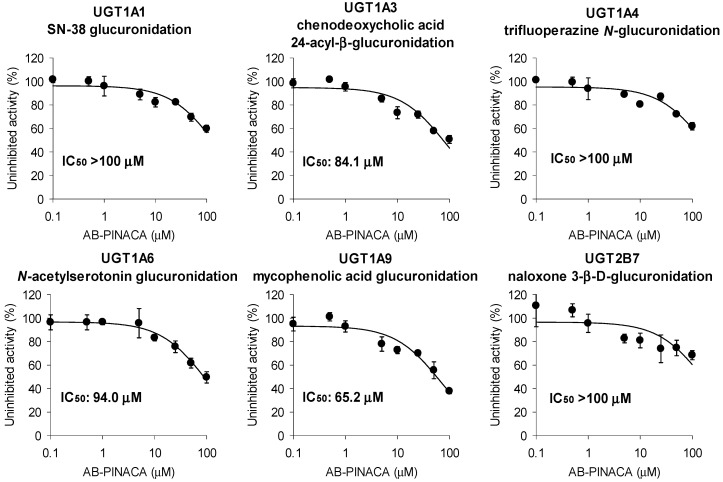
Inhibitory effects of AB-PINACA on metabolic activities of UGT1A1, UGT1A3, UGT1A4, UGT1A6, UGT1A9, and UGT2B7 in ultrapooled human liver microsomes. The cocktail UGT probes consist of A set (0.5 µM SN-38, 2 µM chenodeoxycholic acid, and 0.5 µM trifluoperazine) and B set (1 µM *N*-acetylserotonin; 0.2 µM mycophenolic acid, and 1 µM naloxone). Data are means ± SD (*n* = 3).

**Figure 5 pharmaceutics-12-01036-f005:**
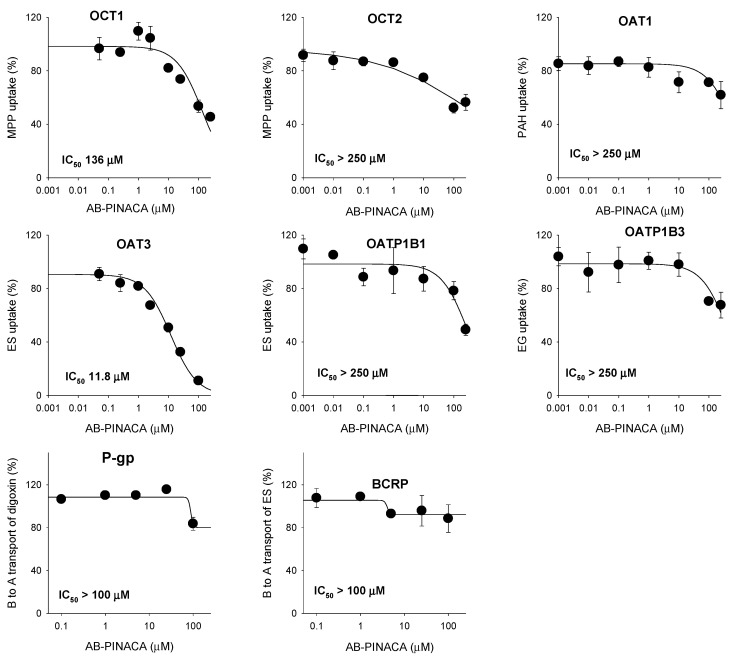
Inhibitory effects of AB-PINACA (0.1–250 µM) on OCT1- and OCT2-mediated ^3^H-methyl-4-phenylpyridinium (MPP^+^) (0.1 μM) uptake, OAT1-mediated ^3^H-para-aminohippuric acid (PAH) (0.1 μM) uptake, OAT3- and OATP1B1-mediated [^3^H]estrone-3-sulfate (ES) (0.1 μM) uptake, OATP1B3-mediated ^3^H-estradiol-17β-d-glucuronide (EG) (0.1 μM) uptake in HEK293 cells overexpressing OCT1, OCT2, OAT1, OAT3, OATP1B1, and OATP1B3. Inhibitory effect of AB-PINACA (0.1–100 µM) on P-gp-mediated B to A transport of ^3^H-digoxin (0.1 μM), and BCRP-mediated B to A transport of ^3^H-ES (0.1 μM) in LLC-PK1-MDR1 and LLC-PK1-BCRP cells. The data are means ± SD (*n* = 3).

**Figure 6 pharmaceutics-12-01036-f006:**
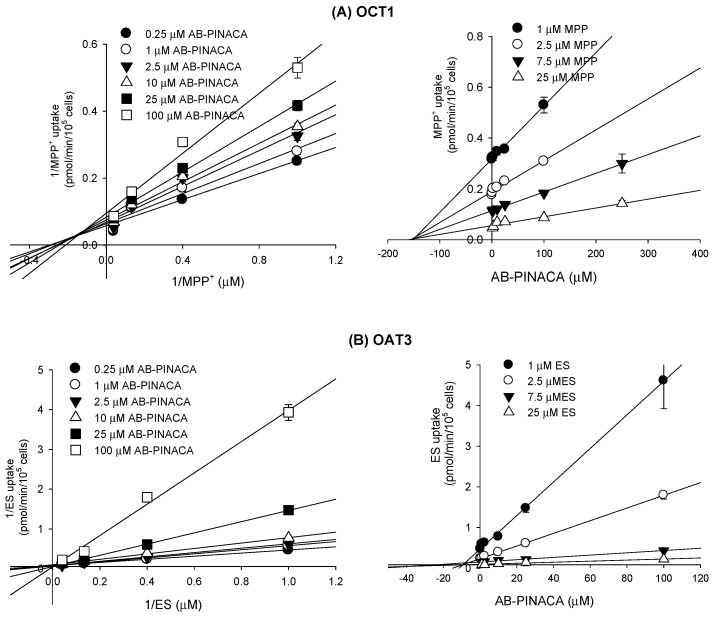
Lineweaver–Burk (**left** panel) and Dixon (**right** panel) plots for the inhibitory effects of AB-PINACA (0.25–100 μM) on (**A**) OCT1-mediated ^3^H-methyl-4-phenylpyridinium (MPP^+^) uptake and (**B**) OAT3-mediated ^3^H-esrone-3-sulfate (ES) uptake in the presence of varying concentrations of MPP^+^ or ^3^H-estrone-3-sulfate (1, 2.5, 7.5, and 25 μM each) in HEK293-OCT1 and HEK293-OAT3 cells, respectively. Each symbol represents the concentration of AB-PINACA. Data are means ± SD (*n* = 3).

**Figure 7 pharmaceutics-12-01036-f007:**
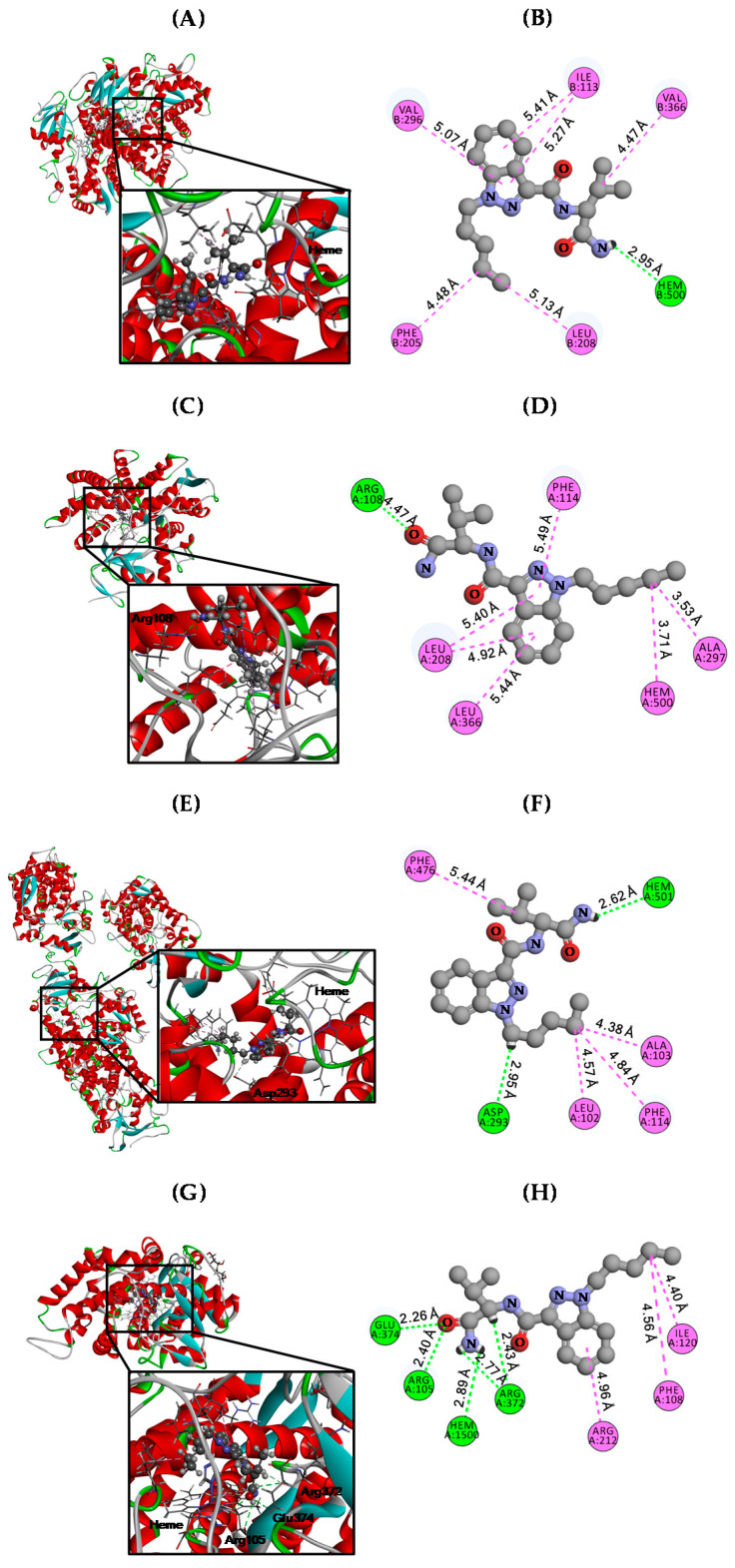
Structural docking analysis of AB-PINACA with CYP2C8 (**A**,**B**), CYP2C9 (**C**,**D**), CYP2C19 (**E**,**F**), and CYP3A4 (**G**,**H**). Mechanisms of molecular action of AB-PINACA with each CYP are shown in 3D structures (**A**,**C**,**E**,**G**) and 2D plots (**B**,**D**,**F**,**H**). The favorable binding of AB-PINACA to the active pocket of each CYP is shown in 3D structures. The hydrogen bonds and hydrophobic interactions between AB-PINACA and each of the CYPs are shown in 2D plots. The docking scores (kcal/mol) are summarized in Table 3. Green, hydrogen bonds; pink, hydrophobic interactions.

**Table 1 pharmaceutics-12-01036-t001:** IC_50_ values of indazole carboxamide synthetic cannabinoid (AB-PINACA) for major cytochrome P450 (CYP) enzyme activities with and without 30-min preincubation with NADPH in ultrapooled human liver microsomes.

CYPs	Enzyme Activities	IC_50_ (μM)		
		Without NADPH Preincubation	With NADPH Preincubation	IC_50_ Shift
CYP1A2	Phenacetin *O*-deethylase	>100	>100	-
CYP2A6	Coumarin 7-hydroxylase	>100	>100	-
CYP2B6	Bupropion hydroxylase	>100	>100	-
CYP2C8	Amodiaquine *N*-deethylase	32.5	30.1	1.1
CYP2C9	Diclofenac 4’-hydroxylase	17.4	19.3	0.9
CYP2C19	[*S*]-Mephenytoin 4’-hydroxylase	20.0	23.0	0.9
CYP2D6	Bufuralol 1’-hydroxylase	>100	>100	-
CYP3A4	Midazolam 1’-hydroxylase	66.4	24.4	2.7

Data represent the average of three measurements. ${\mathrm{IC}}_{50}\;\mathrm{shift}=\;\frac{{\mathrm{IC}}_{50}\;\mathrm{of}\;\mathrm{AB}-\mathrm{PINACA}\;\mathrm{without}\;\mathrm{NADPH}\;\mathrm{preincubation}}{{\mathrm{IC}}_{50}\;\mathrm{of}\;\mathrm{AB}-\mathrm{PINACA}\;\mathrm{with}\;\mathrm{NADPH}\;\mathrm{preincubation}}.$

**Table 2 pharmaceutics-12-01036-t002:** Summary of mode of inhibition and inhibition constant (***K**_i_*) of AB-PINACA on CYPs and transporters.

CYPs and Transporters	***K**_i_* (μM)	Mode of Inhibition
CYP2C8	16.9	mixed inhibition
CYP2C9	16.1	competitive inhibition
CYP2C19	6.7	mixed inhibition
CYP3A4	17.6 (***k**_inact_* 0.04047 min^−1^)	time-dependent inhibition
OCT1	145.7	mixed inhibition
OAT3	8.3	competitive inhibition

***K**_i_* values were calculated from the results in Figure 2, Figure 3 and Figure 6.

**Table 3 pharmaceutics-12-01036-t003:** Interactions between AB-PINACA and human CYP2C8, CYP2C9, CYP2C19, and CYP3A4.

CYPs	CDOCKER Energy (kcal/mol)	Hydrogen Bond (Å)	Hydrophobic (Å)
CYP2C8	−29.36	Heme (4.47)	Ile113 (5.27 & 5.41)
			Phe205 (4.48)
			Leu208 (5.13)
			Val296 (5.07)
			Val366 (4.47)
CYP2C9	−26.99	Arg108 (4.47)	Leu208 (5.40 & 4.92)
			Phe114 (5.49)
			Ala297 (3.53)
			Leu366 (5.44)
			Heme (3.71)
CYP2C19	−33.15	Asp239 (2.95) Heme (2.62)	Leu102 (4.57)
			Aal103 (4.38)
			Phe114 (4.84)
			Phe476 (5.44)
CYP3A4	−32.34	Arg105 (2.40)	Phe108 (4.56) Ile120 (4.40) Arg212 (4.96)
		Arg372 (2.77 & 2.43)	
		Glu374 (2.26)	
		Heme (2.89)	

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
