# Peer review of "Inhibitory Effect of AB-PINACA, Indazole Carboxamide Synthetic Cannabinoid, on Human Major Drug-Metabolizing Enzymes and Transporters"

_pharmaceutics, 2020, doi:10.3390/pharmaceutics12111036_

Round 1

Reviewer 1 Report

The authors investigated the effects of AB-PINACA on CYPs, UGTs, SLCs, MDR1 and BCRP, and its potential drug-drug interactions (DDIs). The authors demonstrated that AB-PINACA:

  • inhibits CYP2C8 (Ki, 16.9 μM), CYP2C9 (Ki, 6.7 μM), and CYP2C19 (Ki, 16.1 μM).
  • inhibits OAT3 (Ki, 8.3 μM)
  • Whether these inhibitions have any clinically meaningful application is unclear as serum AB-PINACA (1.8-127 nM) are lower than the Ki values for the CYPs and transporters abovementioned.
  • did not have significant effects on CYP1A2, CYP2A6, CYP2B6, CYP2D6, UGT1A1, UGT1A3, UGT1A4, UGT1A6, UGT1A9, UGT2B7, OAT1, OATP1B1, OATP1B3, OCT1, OCT2, MDR1 and BCRP
  • exhibits time-dependent inhibition on CYP3A4(Ki, 17.6μ M; kinact, 0.04047 min-1). Inhibitory efficiency ~2.3 ml/micromole/min, which is a low-to-moderate inhibition.

The authors also did Structural docking analysis of AB-PINACA with CYP2C8, 2C9, 2C19 and 3A4.

Overall, the study is descriptive but informative.

Minor issue:
Page 14, line 366, …figure 4 should be figure 3.

Author Response

Minor issue:

Page 14, line 366, …figure 4 should be figure 3.

(Answer) Thank you for the reviewer’s positive comments and we corrected typographical error in the revised manuscript.

Reviewer 2 Report

This paper looks sound, and is of interest to the journal’s readership. However, there are some issues that need to be further explored by the authors, and therefore the paper needs to be improved before publication in Pharmaceutics can be considered. I think it is acceptable after moderate modifications.

The authors already have experience with these substances; in fact, they recently published a work in Molecules.

Some information about the pharmacokinetic profile of AB-PINACA is interesting. Please include more information.

More information about the chromatographic analysis should be included. Please include as supplementary material more information about the parameters used for quantification.

In addition, the inclusion of a chromatogram is necessary.

Once the authors have other publications, it would be interesting to compare the behavior of AB-PINACA with that of other synthetic cannabinoids.

Why the authors use a time of 30 min for incubation? Why IC5= values towards CYP2C9, 2C19 AND 2D6 were higher than those without preincubation?

In Figure 2, some values have higher standard deviation; is there any particular reason for obtaining these values?

In figure 4, why didn’t the authors use concentrations higher than 100 µM for the assays?
